# IMPROVING SAMPLE COMPLEXITY WITH OBSERVATIONAL SUPERVISION

**Khaled Saab**
Department of Electrical Engineering
Stanford University
Stanford, CA, 94305
ksaab@stanford.edu

**Jared Dunnmon**
Department of Computer Science
Stanford University
Stanford, CA, 94305
jdunnmon@stanford.edu

**Alex Ratner**
Department of Computer Science
Stanford University
Stanford, CA, 94305
ajratner@stanford.edu

**Daniel Rubin**
Department of Biomedical Data Science
Stanford University
Stanford, CA, 94305
dlrubin@stanford.edu

**Christopher Ré**
Department of Computer Science
Stanford University
Stanford, CA, 94305
chrismre@stanford.edu

## ABSTRACT

Supervised machine learning models for high-value computer vision applications such as medical image classification often require large datasets labeled by domain experts, which are slow to collect, expensive to maintain, and static with respect to changes in the data distribution. In this context, we assess the utility of *observational supervision*, where we take advantage of passively-collected signals such as eye tracking or "gaze" data, to reduce the amount of hand-labeled data needed for model training. Specifically, we leverage gaze information to directly supervise a visual attention layer by penalizing disagreement between the spatial regions the human labeler looked at the longest and those that most heavily influence model output. We present evidence that constraining the model in this way can reduce the number of labeled examples required to achieve a given performance level by as much as 50%, and that gaze information is most helpful on more difficult tasks.

## 1 INTRODUCTION

Medical imaging is a compelling application area for supervised machine learning methods. Convolutional Neural Networks (CNNs) in particular have recently achieved promising results on applications ranging from cancer diagnosis (Esteva et al., 2017) to radiograph worklist prioritization (Dunnmon et al., 2018); however, these results rely on massive hand-labeled datasets. This requirement for large hand-labeled datasets - which are expensive, because they require physician time to create - has hampered efforts to deploy these models to improve clinical outcomes.

To reduce this labeling cost, we explore rich observational signals that can be passively collected at annotation time, such as eye tracking (or "gaze") data, which describes where a person has looked while performing a task (Barrett et al., 2018; Rodríguez et al., 2018). This approach is possible because of recent advances in eye tracking technology, which has quickly transformed from a technique that was intrusive, inaccurate, and expensive into one that is viable for real-time gaze data collection (Fu et al., 2016). Inspired by the success of eye tracking techniques in NLP applications that only use gaze signal at train time (Hollenstein & Zhang, 2019), we examine a straightforward mapping of gaze data to visual attention layer activations in a way that encourages the model to draw influence from the same spatial regions most heavily utilized by the human annotator.

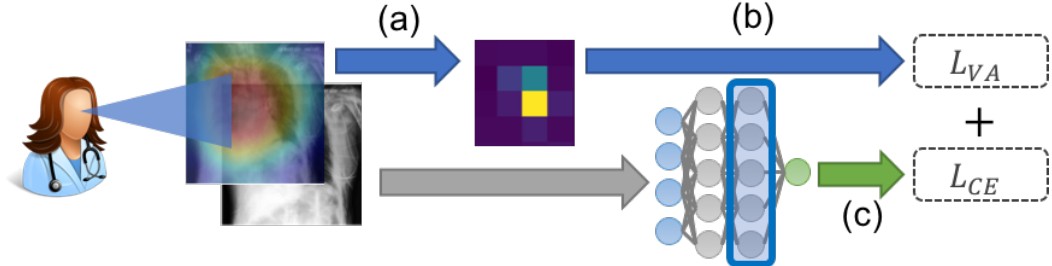

Figure 1: In our approach: (a) gaze signal is passively collected during annotation; (b) this gaze signal is downsampled and used in a visual attention loss, $L_{VA}$; (c) a cross entropy loss, $L_{CE}$, is added to the overall loss.

While noisy observational signals are challenging to extract useful information from, we show that incorporating them alongside traditional declarative labels can reduce the amount of hand-labeled data required to achieve a given level of performance. We first apply our proposed technique to a simple image classification task, where we show that we can maintain model performance using as few as 50% of the training images when gaze information is incorporated at training time. We then examine how the difficulty of the task impacts this labeled data reduction, and show that observational signals appear to be more helpful for difficult tasks. While we demonstrate our approach on a non-medical dataset for which gaze data was available, similar reductions in required labeled data, particularly for difficult tasks, could improve the feasibility of training useful models for medical imaging tasks.

## 2 RELATED WORK

The work presented here draws inspiration from recent research on weak supervision, the use of gaze data in machine learning systems, and attention-based models.

Weak supervision has become an increasingly popular method for training machine learning models with limited hand-labeled data. Indeed, methods such as distant supervision (Alfonseca et al., 2012), data programming (Ratner et al., 2016; 2017), and crowdsourcing (Yuen et al., 2011) have shown substantial success in decreasing the amount of labeling time required to train high-performance machine learning models. Nonetheless, these approaches require supervision signals that are fundamentally declarative – that is, they do not leverage the rich information contained in user behavior that can be captured by passive sensing modalities such as eye tracking and click stream monitoring.

Of particular relevance for this work are those studies that have examined using gaze data directly in the context of training computer vision models. Karessli et al. (2017) collect eye tracking data on several different datasets and demonstrate that features derived from this data can support class-discriminative representations that improve zero-shot fine-grained image classification; notably, this approach requires gaze data at test time. Wang et al. (2017) integrate gaze data directly into the optimization procedure for a modified Latent Support Vector Machine (LSVM) and demonstrate competitive performance on several image classification and localization tasks in a manner that is gaze-free at test-time. While conceptually similar, the modeling and optimization approach pursued by Wang et al. (2017) is substantially different than that explored here.

Integration of visual attention techniques into computer vision algorithms has become a direction of significant research interest. An *attention mechanism* represents a learned weighting applied to different subsamples of a given input (e.g. image patches, sequence elements, etc.). Recent efforts to directly incorporate learned visual attention layers into the CNN training process have continually led to state-of-the-art performance on tasks including image classification (Rodríguez et al., 2018), object recognition (Ba et al., 2014), visual question answering (Yang et al., 2016; Johnson et al., 2017), and medical image segmentation (Oktay et al., 2018). In our work, we leverage the same concept of visual attention that has enabled these advances as a fundamental primitive for gaze-based supervision.

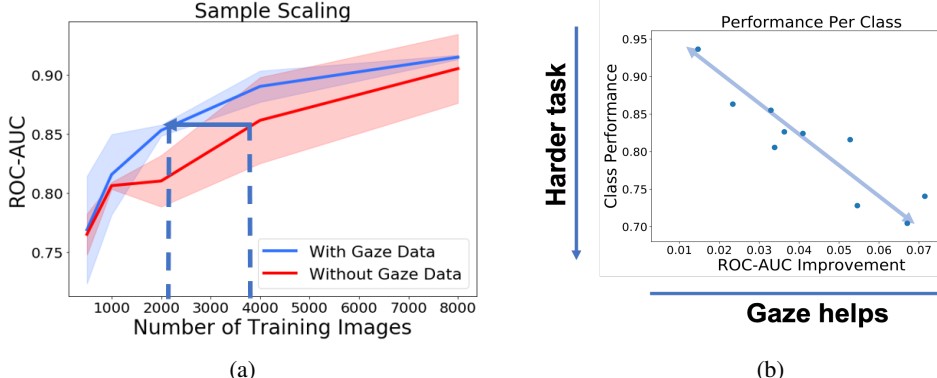

Figure 2: (a) Mean ROC-AUC across 10 classes for the POET image classification task versus number of training images (shaded regions are standard deviation). We observe superior performance at all training set sizes when integrating gaze data. (b) Per-class performance for models trained on 2000 examples versus ROC-AUC improvement when using gaze data. The strong negative correlation indicates that gaze data is more helpful as the classification becomes more difficult.

## 3 METHODS

While much initial work on the use of eye tracking signal has considered using the gaze signal as another set of features for the model, we are motivated by the prospect of using it to train standard neural network architectures that do not need gaze features at test time, and are thus practically deployable in settings where eye trackers are unavailable. To this end, we integrate the eye tracking signal into the training loss function by balancing the classification loss with a second loss term describing the deviation between model and human visual attention as shown in Fig. 1,

$$
\begin{aligned}
L &= L_{CE} + \alpha L_{VA} + \lambda R_{\ell_2} \\
&= \sum_{i=0}^{N} \left[ -\sum_{j=0}^{C} y_{i,j} \log(p_{i,j}) + \alpha MSE(\tilde{H}_i, \tilde{M}_i) \right] + \lambda ||\theta||_{\ell_2},
\end{aligned}
\tag{1}
$$

with $L_{CE}$ the cross-entropy loss, $L_{VA}$ the visual attention loss, $R_{\ell_2}$ the regularization term, $p_{i,j}$ the probability of observation $i$ out of $N$ belonging to class $j$ out of $C$, $y_{i,j}$ an indicator for the true label, $\lambda$, $\alpha$ constant hyperparameters, $\theta$ the set of model parameters, $MSE$ the mean squared error, $\tilde{M}$ the $\ell_1$-normalized Class Activation Map (CAM, $M$) of Zhou et al. (2016), and $\tilde{H}$ the $\ell_1$-normalized human attention map $H$ obtained by integrating the eye tracking signal in time. For a fully convolutional network wherein the field of view of any given convolutional filter can be determined via simple upsampling, $M$ can be directly calculated as,

$$
M(x,y) = \sum_k w_k^{c^*} f_k(x,y),
\tag{2}
$$

where $w_k^{c^*}$ is the weight corresponding to the true class label for each example and $f_k(x,y)$ represents the activation of unit $k$ in the last convolutional layer at spatial location $(x,y)$. Because CAMs generated by CNNs give signals at a resolution substantially lower than that of the original image (e.g. for an 18-layer residual network operating on a 224 x 224 input image, the CAM is 7 x 7), we downsample $H$ such that its resolution is aligned with that of $M$.

## 4 EXPERIMENTS

We hypothesize that the additional supervision signal provided by gaze data can enable model training procedures that reduce the amount of training data required to achieve a given level of performance, especially for difficult tasks. To test this hypothesis, we assess whether training a simple 3-layer CNN with gaze data improves classification performance on the public Pascal Objects Eye Tracking (POET) dataset (Papadopoulos et al., 2014), as measured by the average Area Under the Receiver

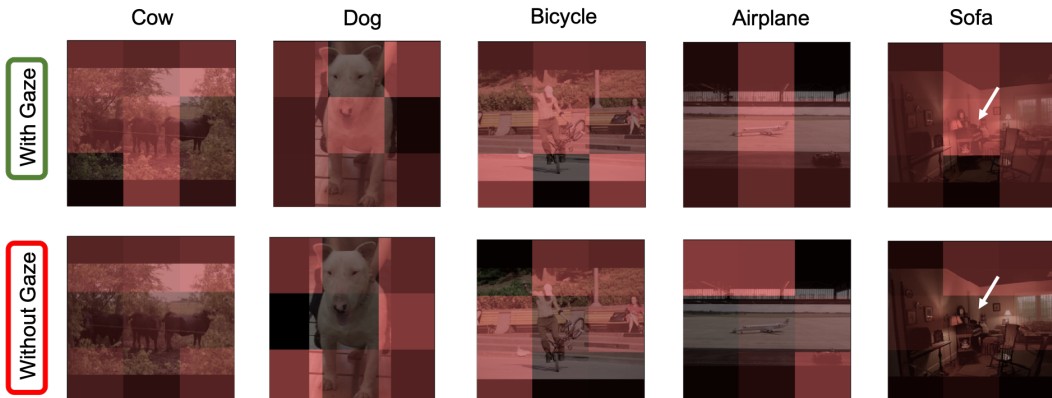

Figure 3: Randomly selected images and CAMs (in red) where the model with gaze achieves the correct classification, but the model without gaze does not. Compared to the model trained without gaze data, the model with gaze data is influenced more by qualitatively meaningful image regions.

Operating Characteristic Curve (ROC-AUC) over runs with five different random seeds. The POET dataset contains 6,270 images spanning 10 classes, along with eye tracking data from five different labelers for each image, which allows us to perform this comparison. Models are trained both with and without gaze data using the standard procedure described in Appendix A. For simplicity, we refer to networks trained with gaze data as "gaze-augmented" networks and those trained without gaze data as "standard" networks. Using this provided gaze data, we construct a heatmap by additively placing 2D Gaussians with means equal to the fixation points (i.e. locations where the labelers maintained visual gaze) and a predefined standard deviation, similar to what is done in eye-tracking toolkits (Dalmaijer et al., 2014).

## 4.1 LABELED DATA REDUCTION

We first analyze the relative sample complexity of the gaze-augmented network and the standard network by using varying numbers of training images. If our hypothesis is correct, the gaze-augmented network will achieve equivalent performance with a smaller number of samples. As seen in Fig. 2a, integrating the gaze-based loss into our training procedure yields superior model performance at every training set size. Further, we find that the gaze-augmented models can achieve similar performance levels to the standard models while using as little as 50% the amount of training data – we observe an average value of 0.85 ROC-AUC for 2,000 images with gaze data versus an average value of 0.86 ROC-AUC for 4,000 images without gaze data. However, the gaze data do not improve performance as much in the high (8000 images) and low (1000 images) data regimes, as the model either does not have enough training examples to learn useful features in the low data regime, or has sufficient training examples such that the gaze data does not provide a performance gain in the high data regime. We also observe a substantial decrease in model variance across different random seeds when using the gaze data (Appendix B); this trend is consistent with the fact that gaze data provides additional constraints on the model parameter space during optimization.

## 4.2 TASK DIFFICULTY

To further understand the circumstances in which observational signals may benefit model training, we examine how the difficulty of a task relates to the usefulness of gaze data at training time. We use per-class performance achieved by the standard network as a proxy for task difficulty, and evaluate the relationship between this quantity and the per-class performance gains when using the gaze-augmented model (Fig. 2b). First, we find that some classes show substantial improvement when using gaze data (7 points ROC-AUC), while others do not (1 point ROC-AUC). Second, we observe a clear negative correlation between the task difficulty and the performance gains from using gaze data, with a Pearson correlation coefficient of $-0.85$ (p-value 0.002). We speculate that these findings may result from the relative difficulty of identifying the different classes, with gaze data providing more benefit for more difficult classes. A more detailed analysis of performance improvements for each class as a function of sample size can be found in Appendix C.

Finally, we qualitatively evaluate changes in model spatial attention when gaze data is added; we expect that gaze-augmented models will have high activations in spatial regions of the image that would be deemed relevant by a human labeler. In Fig. 3 we show a few random cases where the gaze-augmented network achieved the correct classification, while the standard network returned the incorrect classification. In contrast to the standard network, we observe that the gaze-augmented network is heavily influenced by qualitatively important parts of the image. These results suggest that constraining spatial activations with gaze data improves classification peformance by ensuring that the model pays attention to relevant parts of the image.

## 5    CONCLUSION

In this study, we introduce a simple method for incorporating observational, passively collected gaze signals into CNN training procedures. We have demonstrated that constraining the model attention to spatial regions deemed relevant to an expert labeler can reduce the amount of labeled data needed to achieve a given level of performance. Additionally, we find that the performance gains from incorporating the observational signal are larger for more difficult tasks. Fully characterizing the circumstances in which we see such gains from gaze-augmented models is a promising avenue for future work. Going forward, we plan to assess the applicability of this technique to medical imaging tasks, and to further investigate how observational signals may improve model robustness.

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

## A    MODEL ARCHITECTURE AND TRAINING DETAILS

In this work, we train simple three-layer CNNs, where each layer has 32 kernels of size 5x5, a stride of 2, zero padding of 2, and rectified linear activation functions. The convolution layers are followed by an average pool layer with a kernel size of 3x3, and finally a fully connected layer with a softmax activation function. The dataset was divided into 70-20-10 train-dev-test splits; however, to improve performance we over-sample the training images such that each class has roughly the same number of examples, resulting in 8,000 images per epoch of training when the full dataset is used. For model training we use the Adam optimizer with $\beta_1 = 0.9$ and $\beta_2 = 0.999$ for 30 epochs while halving the learning rate upon a two-epoch plateau in the validation accuracy. We then cross-validate across five random train-dev splits using different random seeds and report mean results. Random hyperparameter search was performed for each combination of $\{\alpha = 0, \alpha \neq 0\}$ over the appropriate subset of $\alpha$: $[10^{-8}, 10^{-1}]$, $\lambda$: $[10^{-8}, 10^{-2}]$, and learning rate: $[10^{-6}, 10^{-2}]$, over 20 trials.

## B    QUANTITATIVE RESULTS

In Table 1, we present quantitative data describing mean model performance across five random seeds as a function of training set size for both gaze-augmented and standard CNNs.

| Sample Size | Gaze-augmented (mean) | Gaze-augmented (std) | Standard (mean) | Standard (std) |
|---|---|---|---|---|
| 1000 | **0.82** | 0.034 | 0.81 | **0.0029** |
| 2000 | **0.85** | **0.0047** | 0.81 | 0.022 |
| 4000 | **0.89** | **0.013** | 0.86 | 0.036 |
| 8000 | **0.91** | **0.0019** | 0.91 | 0.029 |

Table 1: Quantitative results describing mean and standard deviation of gaze-augmented and standard models with increasing training set size.

## C    PER CLASS PERFORMANCE ANALYSIS

In Fig. 4, we show the model improvement for each class and training set size. Taking cow versus aeroplane as an example, inspection of the dataset demonstrates many images with the label cow usually have multiple cows in them, and the cows can greatly vary in appearance; for instance, some cows have horns (i.e. are bulls), some are very skinny, and some only partially appear in the image. Conversely, most images with the label aeroplane only include one aeroplane near the center of the image, and have distinctive and consistent features, e.g. sharp edges for the wings and cone shaped nose. We suspect that for these reasons, the gaze data provided for classifying more difficult classes, such as cow, may be more helpful than those provided for identifying easier classes, such as aeroplane.

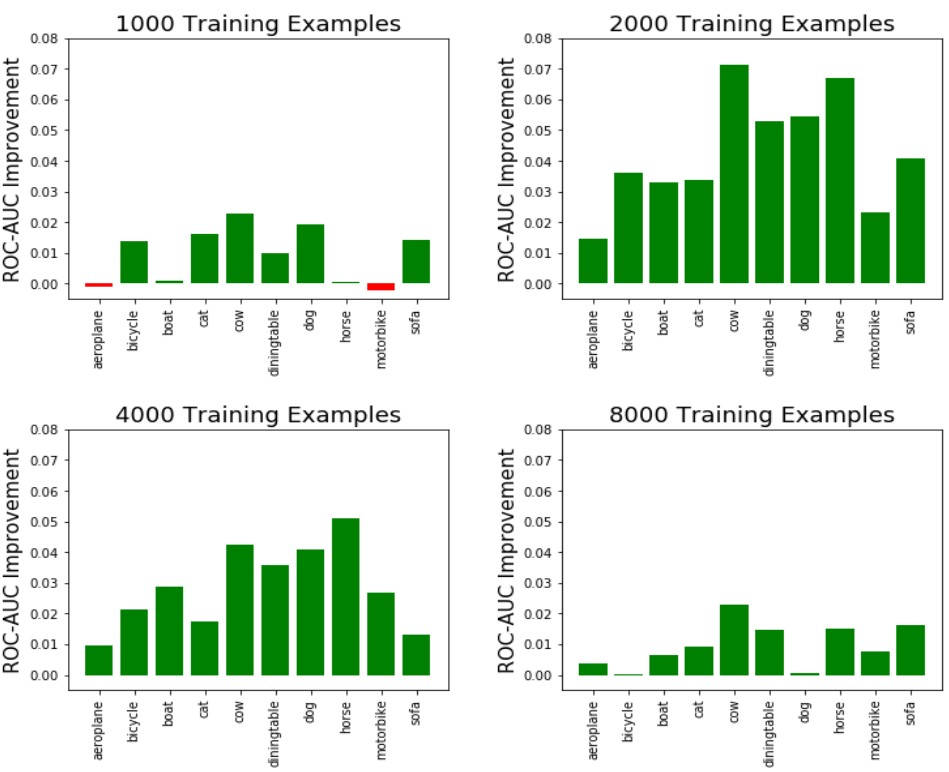

Figure 4: Mean ROC-AUC improvement from training wth gaze data for each class at multiple different training set sizes. Green indicates improvement, red indicates degradation.

