# OpenReview forum: "Improving Sample Complexity with Observational Supervision"
_ICLR.cc/2019/Workshop/LLD — LLD 2019_

### Official Review · AnonReviewer1 · 2019-04-07
**Interesting idea with encouraging initial results**

**Rating:** 4
**Confidence:** 2

**Review:**

This work proposed to use gaze information in order to reduce the sample complexity of a model and the needed labeling effort to get a target performance. The proposed method uses an attention layer and adds a penalty to reduce the gap between downsampled human attention maps and class activation maps. The experimental results show an improvement especially for middle sized samples, and a higher effect for harder tasks.

The idea and method is overall interesting, and would be interesting to discuss. One direction to build on this work, other than extending the experiments to more complex, larger scale applications, is to try to leverage more information from the human attention maps, as downsampling throws away some of the information that can be interesting for training.

---

### Official Review · AnonReviewer2 · 2019-04-08
**Simple idea to incorporate gaze data into standard CNN architectures for image classification.**

**Rating:** 4
**Confidence:** 2

**Review:**

This paper presents a simple method to incorporate gaze signals into standard CNNs for image classification, adding an extra term in the loss function. The term is based in the difference between the Class Activation Map obtained from the model, and the human map constructed using the eye tracking information. The authors apply their method to the POET dataset and report interesting results when using different sizes for the training set. They show that the gazed network achieved equivalent performance to that of a standard CNN using less training data for intermediate data regimes.

The paper well written. It presents a simple idea which has a lot of potential, specially in the context of medical data (as suggested by the authors in their planned future works). Some comments I would like to see in the camera ready version of this work:

- It is not clear how the human attention map is constructed. The authors just say that this is obtained by “integrating the eye tracking signal in time”. Since this is a crucial element in their framework, I would like to see a detailed description of how this is obtained. If space constraint is a problem, you could just add an appendix section with this info.
- In the orange line in Figure 2 (the line associated to the standard CNN) I do not see the std. This value is reported in table 2 (Appendix B), so I guess this can be a problem related to image transparency. Please, fix this problem so that we can see the confidence interval for the standard CNN as we can do with the gazed CNN.
- Do you think this idea could be also useful to improve image segmentation based on CNNs with limited data?

Minor corrections:
- In Section 4: “To test the this hypothesis” should be “To test this hypothesis”.

---

### Decision · Program_Chairs · 2019-04-08
**Acceptance Decision**

Accept